# Fenugreek (*Trigonella foenum-graecum* L.) Seeds Dietary Supplementation Regulates Liver Antioxidant Defense Systems in Aging Mice

**DOI:** 10.3390/nu12092552

**Published:** 2020-08-24

**Authors:** Devesh Tewari, Artur Jóźwik, Małgorzata Łysek-Gładysińska, Weronika Grzybek, Wioletta Adamus-Białek, Jacek Bicki, Nina Strzałkowska, Agnieszka Kamińska, Olaf K. Horbańczuk, Atanas G. Atanasov

**Affiliations:** 1Department of Pharmacognosy, School of Pharmaceutical Sciences, Lovely Professional University, Phagwara, Punjab 144411, India; dtewari3@gmail.com; 2Institute of Genetics and Animal Biotechnology Polish Academy of Sciences, 05-552 Magdalenka, Poland; w.grzybek@ighz.pl (W.G.); n.strzalkowska@ighz.pl (N.S.); 3Division of Medical Biology, Institute of Biology, Jan Kochanowski University, Uniwersytecka 7, 25-406 Kielce, Poland; malgorzata.lysek-gladysinska@ujk.edu.pl; 4Department of Surgical Medicine with the Laboratory of Medical Genetics, Collegium Medicum, Jan Kochanowski University, al. IX Wieków Kielc 19A, 25-317 Kielce, Poland; wioletta.adamus-bialek@ujk.edu.pl (W.A.-B.); jacek.bicki@ujk.edu.pl (J.B.); 5Faculty of Medicine, Collegium Medicum, Cardinal Stefan Wyszynski University, 01-815 Warsaw, Poland; agnieszka.kaminska73@wp.pl; 6Department of Technique and Food Product Development, Warsaw University of Life Sciences (WULS-SGGW) 159c Nowoursynowska, 02-776 Warsaw, Poland; olaf_horbanczuk@sggw.pl; 7Institute of Neurobiology, Bulgarian Academy of Sciences, 23 Acad. G. Bonchev str., 1113 Sofia, Bulgaria; 8Department of Pharmacognosy, University of Vienna, 1090 Vienna, Austria; 9Ludwig Boltzmann Institute for Digital Health and Patient Safety, Medical University of Vienna, Spitalgasse 23, 1090 Vienna, Austria

**Keywords:** fenugreek, aging, liver, antioxidants, mice

## Abstract

Fenugreek seeds are widely used in Asia and other places of the world for their nutritive and medicinal properties. In Asia, fenugreek seeds are also recommended for geriatric populations. Here, we evaluated for the first time the effect of fenugreek seed feed supplementation on the liver antioxidant defense systems in aging mice. The study was conducted on 12-months aged mice which were given fenugreek seed dietary supplement. We evaluated the activities of various antioxidant defense enzymes such as superoxide dismutase (SOD), glutathione reductase (GR), and glutathione peroxidase (GPx), and also estimated the phenolics and free radical scavenging properties in mice liver upon fenugreek supplementation. The estimation of SOD, GPx, and GR activities in aged mice liver revealed a significant (*p* < 0.01) difference among all the liver enzymes. Overall, this study reveals that fenugreek seed dietary supplementation has a positive effect on the activities of the hepatic antioxidant defense enzymes in the aged mice.

## 1. Introduction

Aging is a syndrome of changes that are deleterious, progressive, universal, and thus far irreversible. Aging changes occur to molecules, to cells, and to organs. With aging, accumulation of toxic material in different tissues is observed. Aging is an aggregation of destruction or impairment of various cells, molecules, and tissues over a lifetime, which mostly leads to malfunction and frailty. Apart from other processes, old age is considered as the biggest risk factor for several diseases conditions, including cardiovascular, neurodegenerative diseases, and cancer [1]. For explanation of ageing process, over 300 theories have been proposed [2], however, none out of these is uniformly accepted by gerontologists. Notably, Denham Harman’s initial theory of the involvement and association of free radicals to the fundamental aging process [2] received emerging acceptance as a plausible explanation of the chemical reactions for ageing process [3].

The liver is a vital organ of the body that is often attacked by reactive oxygen species (ROS) [4]. ROS are produced under naturally occurring physiological conditions and significantly contribute in the development of numerous human ailments. A range of antioxidant systems are implied in defense mechanisms of cells against ROS, which can be produced in excess under conditions of stress, and these antioxidant systems include both non-enzymatic and enzymes antioxidants [5]. It has been reported that diet rich in antioxidants enhances the cellular defense mechanisms, by decreasing ROS levels that are generated through cell metabolism in normal cellular conditions [6]. Nicotinamide adenine dinucleotide phosphate (NADPH) is a potent reducing agent that is required to maintain antioxidants of cells in a reduced state, including glutathione [7].

It is reported that increased levels of apoptosis and oxidative damage are presented in the liver of aged mammals. Mainly sinusoidal endothelial cells and bile duct cells are the apoptotic cells in aged livers, since these cell types are highly sensitive to injury by oxidative stress. Therefore, it was assumed that ageing of the liver is mainly induced by the apoptosis induced by ROS in these cells [8]. Recently, rodent studies revealed that there is an involvement of both extracellular and intracellular factors in the liver mass recovery impairment during aging. Among the intracellular factors, an age-dependent decrease of Sirtuin-1 (SIRT1), budding uninhibited by benzimidazole-related 1 (BubR1), and Yes-associated protein (YAP) is related to tissue reconstitution dampening and also to cell cycle genes inhibition following partial hepatectomy. Thus, it is also well-known that the capacity of liver regeneration after resection decreased with aging [9].

Fenugreek, botanically equated with *Trigonellafoenum-graecum* L., is a leguminous herb of the Fabaceae family. The seeds of this plant are widely used in Asia, Mediterranean countries, and Africa, as an ingredient of regular diets [10], and are also utilized as medicine, fragrances, in cosmetics, beverages, and for industrial purposes [11]. The seed of fenugreek has a yellow and hard embryo which is covered through a larger and corneous layer semi-transparent white endosperm. Various phytochemicals are present in fenugreek like steroidal sapogenins. The oily embryo of the fenugreek contains diosgenin, a well-known steroidal precursor. The endosperm has saponin and protein content and its husk found to have higher total polyphenols [12,13]. Fenugreek seeds have been found to contain several coumarin compounds as well as a number of alkaloids (e.g., trigonelline, gentianine, carpaine). The major bioactive compounds in fenugreek seeds are believed to be polyphenol compounds, such as rhaponticin and isovitexin [12,14].

A small amount of volatile oils and fixed oil has been found in fenugreek seeds [15]. Some important compounds present in the volatile oil of fenugreek are diacetyl, 1-Octene-3-one, sotolon, acetic acid; eugenol, butanoic acid, caproic acid, 3-Isobutyl-2-methoxypyrazine, isovaleric acid, 3-isopropyl-2-methoxypyrazine, 3-Amino-4,5-dimethyl-3, linalool, (Z)-1,5-Octadiene-3-one, and 4-dihydro-2(5H)-Furanone [12]. Some of the important phytoconstituents of fenugreek seeds are presented in Figure 1.

Various pharmacological effects are attributed to this plant such as antiviral, antimicrobial, carminative, anticholesterolemic, febrifuge, restorative, laxative, expectoral, galactogogue, uterine tonic, anticarcinogenic, anti-inflammatory, antioxidant, hypotensive, etc., [16,17]. Fenugreek seeds are used in traditional medicine as an antidiabetic, gastric stimulant and also utilized against anorexia. In recent decades, many health benefits of fenugreek seeds were clinically and preclinically validated [18].

Here, we evaluated the effect of fenugreek seed feed supplementation on the liver antioxidant defense systems in aging mice. To the best of our knowledge, this is the first study that investigated the effect of fenugreek seed dietary supplementation in this context.

## 2. Materials and Methods

### 2.1. Chemicals and Seed Supplement

Superoxide Dismutase Assay Kit, Item No. 706,002 (Cayman Chemical Company; Ann Arbor, Michigan 48,108 USA), Glutathione Peroxidase Assay Kit, Item No. 703,102 (Cayman Chemical Company; Ann Arbor, Michigan 48,108 USA), microplate reader Synergy4 (Biotek; Winooski, VT 05404, USA), Glutathione Peroxidase Assay Kit, Item No. 703,202 (Cayman Chemical Company; Ann Arbor, MI 48108, USA), vitamin C (A92902 Sigma-Aldrich L-Ascorbic acid 99%). The fenugreek seeds were purchased from the local pharmacy Herbapol in Cracow SA, Polish herbs company (Nr. of claims GIS-HŻ-4433-D-168/AW/05, GTIN: 5903850001270). The seeds were kept in the original manufacturer’s packaging in a refrigerator (in temperature 5 °C +/− 1.5 °C).

### 2.2. Experimental Animals

The study was carried out on twenty-four Swiss 12-months old male mice from the population maintained at the Institute of Genetics and Animal Breeding, the Polish Academy of Sciences in Jastrzębiec. The animals were maintained in standard cages of the farm at temperature 22 °C under standard conditions with 12 h of daylight and 12 h of darkness, with access to food and water. All of the described experiments were approved by the Local Ethical Commission. All procedures were performed according to the guiding principles for the care and use of research animals and were approved by the Bioethical Commission operating at the Świętokrzyska Lakewood Chamber, Kielce, Poland [No. 46/2016].

Throughout the experiment, the animals were fed with feed dedicated to breeding mice with a total energy of 3625.85 kcal/kg (the detailed composition of the feed is listed in Table 1).

The animals were divided into four groups (*n* = 6) two controls (Control Start, CS and Control End, CE) and experimental (Ex5 and Ex10). The needed sample size for this animal study was determined according to the resource equation approach, using the free web tool available at http://wnarifin.github.io. The animals were kept in cages of three mice each. Before feeding fenugreek, antioxidant status was determined in the liver of control mice (Control Start). The experimental animals (Ex5 and Ex10) for 4 weeks received a diet additionally, whole fenugreek seeds 0.125 g and 0.250 g per 2.5 g of feed, respectively (Figure 2). There were no differences in the amount of feed intake among the groups and the daily consumption (g per box) from cage was: CS = 6.92 ± 0.23; CE = 6.89 ± 0.17; Ex5 = 6.93 ± 0.21; Ex10 = 6.91 ± 0.21.

After 4 weeks, the antioxidant potential in the liver of all was determined, using the same analytical procedures. Four hours after the final treatment all animals were anesthetized with diethyl ether and then decapitated. Immediately after decapitation, the liver was isolated and processed for biochemical studies. There were no statistically significant differences in the body weights of the animals of the different groups, which were as follows: CS = 33.7 ± 2.3; CE = 34.1 ± 1.9; Ex5 = 34.2 ± 2.4; Ex10 = 33.9 ± 2.1. We also did not observe any histopathological or morphological changes in the livers of the mice of the different groups, and the weights of livers among the different groups also did not significantly vary: CS = 1.83 ± 0.19; CE = 1.81 ± 0.21; Ex5 = 1.82 ± 0.24; Ex10 = 1.79 ± 019. The further procedure strictly followed instructions of the producer.

### 2.3. Estimation of Superoxide Dismutase (SOD)

Perfusion of liver tissue was made in phosphate buffered-saline (PBS), at pH 7.4. 1 g of liver tissue was homogenized in 5 mL of a 20 mM 4-(2-hydroxyethyl)-1-piperazineethanesulfonic acid (HEPES) buffer (pH 7.2, 1 mM ethylenediaminetetraacetic acid (EDTA), 210 mM mannitol, 70 mM sucrose, per 1 g of tissue) chilled to 4 °C. After that, the obtained homogenates were centrifuged at 2500× *g* for 15 min at 4 °C. To prevent uncontrolled reaction initiation, in this assay it is crucial to store samples on ice until the analysis will be started. The assay procedure was conducted by using the Superoxide Dismutase Assay Kit, Item No. 706,002 (Cayman Chemical Company; Ann Arbor, MI 48108 USA). Absorbance was measured in duplicate with the help of a microplate reader Synergy4 (Biotek; Winooski, VT 05404, USA). The total activity of superoxide dismutase was expressed in U/mL.

### 2.4. Estimation of Glutathione Peroxidase (GPx)

Perfusion of liver tissue was made in PBS buffer, at pH 7.4. Homogenization of 1 g liver tissue sample was executed in 5 mL of buffer containing 50 mM Tris-HCl, 5 mM EDTA, and 1 mM dithiothreitol. Homogenates were centrifuged at 10,000× *g* for 15 min at 4 °C. Supernatants were placed on ice until the analysis started. The procedure was conducted according to the manufacturer instructions by using the Glutathione Peroxidase Assay Kit, Item No. 703,102 (Cayman Chemical Company; Ann Arbor, MI 48108, USA). Measurements of reaction kinetics in triplicate were made with the help of a microplate reader Synergy4 (Biotek; Winooski, VT 05404, USA). Glutathione peroxidase activity was expressed in nmol/min/mL.

### 2.5. Estimation of Glutathione Reductase (GR)

Perfusion of liver tissue was made in PBS, at pH 7.4. Total of 1 g of liver tissue was homogenized in 5 mL of could buffer 50 mM potassium phosphate (pH 7.5) with 1mM EDTA chilled to 4 °C. After that, the obtained homogenates were centrifuged at 10 000× *g* for 15 min at 4 °C. The procedure was conducted according to the manufacturer instructions by using the Glutathione Peroxidase Assay Kit, Item No. 703,202 (Cayman Chemical Company; Ann Arbor, MI 48108, USA). Measurements of reaction kinetics in triplicate were made with the help of a microplate reader Synergy4 (Biotek; Winooski, VT 05404, USA). Glutathione reductase activity was expressed in nmol/min/mL.

### 2.6. Vitamin C

The level of vitamin C in livers was determined using a LambdaBio-20 spectrophotometer (Perkin Elmer, Waltham, MA, USA). Total of 0.5 mL of tissue homogenate, 0.5 mL of distilled water and, 1.0 mL of 10% TCA were mixed thoroughly and centrifuged for 20 min. Total of 1.0 mL of the obtained supernatant was combined with 0.2 mL of 2,4dinitrophenylhydrazine–thiourea–copper sulphate reagent and incubated at 37 °C for 2 h. Subsequently, 1.5 mL of 65% sulfuric acid was added and mixed, and the sample remained at room temperature for another 30 min. Change in color of the sample was measured at 520 nm. The solutions of vitamin C standards (0.5–5 mg of vitamin C A92902 Sigma-Aldrich L-Ascorbic acid 99%) were treated similarly.

### 2.7. DPPH

Measurements for radical scavenging activity were performed with a routine assay procedure [19] using a synthetic DPPH radical (1,1-diphenyl-2-picrylhydrazyl). Folin-Ciocalteu reagent was used as an oxidizing reagent and all the chemicals were purchased from Sigma-Aldrich ChemieGmbh (Munich, Germany) in the highest available purity.

### 2.8. Total Phenols

The determination of the content of total phenols was performed as previously described [20]. The samples were thoroughly mixed, and after 8 min, 2 mL of the saturated sodium carbonate solution was added. The next stage of the analysis involved the incubation test at 40 °C for 30 min (until a stable characteristic blue color was developed). The absorbance was measured at 765 and 735 nm against a blank sample (experimental material replaced with 0.5 mL ddH2O). Results were read using a calibration curve plotted based on the absorbance of the gallic acid standard in the range of 0 to 0.5 mg/mL and expressed in mg of GAE/g tissue or mg of GAE/mL serum (GAE—gallic acid equivalent).

### 2.9. Malondialdehyde Level

The muscle tissue was perfused with a phosphate buffer, at pH 7.4. Afterward, it was homogenized in 2 mL of a phosphate buffer, chilled to 4 °C, with the addition of 20 µL of butylated hydroxytoluene in acetonitrile. The samples were centrifuged at 10,000× *g* for 10 min at a temperature of 4 °C. The obtained supernatant was stored on ice until the analysis. The further procedure strictly followed instructions of the producer of the OxisReasearch™ Bioxytech^®^ MDA-586 ™ test (Foster City, CA 94404-1136, USA). The absorbance (λ586) was read using the Cary Varian 50Bio spectrometer (Santa Clara, CA 95051, USA). Calculations were performed on the basis of a calibration curve obtained according to the producer’s recommendations and the template included in the test report. The MDA level was expressed in µM.

### 2.10. Statistical Analysis

The results were subjected to statistical treatment with the use of statistical software IBM SPSS 18, using a general linear model (GLM). The conducted multi-way (age, feeding) analysis of variance ANOVA enabled the identification of significant effects of differences at the level of *p* ≤ 0.01. The parameters of dependent variables were measured on the quantitative scale, and their distributions were similar to normal distributions. Statistical tests (Pillai trace, Wilks’ lambda, Hotelling’s trace, largest Roy’s element) were applied. In addition, multiple pairwise comparisons were performed to analyze the difference between each pair. The Bonferroni method was used, which through a *t*-test controls the level of the overall error by setting an error rate for each test to be equal to the level of the experimental error divided by the number of tests.

## 3. Results

The estimation of SOD, GPx, and GR in aged mice liver showed that there was a significant (*p* < 0.01) difference observed among all the liver enzymes. The concentration of SOD was found less in the CS (21.97 ± 0.32) and CE (21.91 ± 0.6) group as compared to the EX5 (30.64 ± 1.04) and EX 10 (35.87 ± 1.61) group. There was no significant variation observed in the amount of SOD in the beginning and the end of the experiment in the control group, however, the fenugreek seed supplementation in a dose-dependent manner increased the SOD in aged mice liver. This clearly shows that fenugreek seed dietary supplementation significantly increased the concentration of the SOD in the aged mice liver.

On the other hand, the levels of GPx and GR1 decreased significantly (*p* < 0.01) with the fenugreek seed dietary supplementation. Similar to that of SOD levels, no significant change was recorded in the CS and CE groups in both GPx and GR1 levels which clearly shows that the levels of GPx and GR1 were almost unchanged after four weeks of the experiment. However, there was a significant decrease in the levels of GPx and GR1 enzyme levels in the aged mice liver which clearly shows that the fenugreek seeds dietary supplement substantially decreases the GPx and GR1 level in a dose-dependent manner. The details of the estimated enzymes are presented in Table 2.

We have also evaluated the free radical scavenging activity, vitamin C, and polyphenolic concentration in mice liver. The analysis showed that although there was a slight increase was recorded in vitamin C concentration in the fenugreek dietary supplement-treated aged mice still, the difference was not significant. Moreover, there was a significant (*p* < 0.05) difference recorded in total polyphenols and DPPH free radical scavenging activity. The EX5% and EX10% group clearly showed a higher concentration of polyphenols and thereafter a higher DPPH free radical scavenging effect was found as compared to the control groups. The details are presented in Table 3.

The fenugreek seed supplementation exhibited tendency to reduce the MDA levels (in both the EX5 and EX10 groups). However, these reduction tendencies did not reach statistical significance [Figure 3].

## 4. Discussion

Many studies have examined the effect of aging on the oxidative status and the injury of livers of old mice associated with oxidative stress [8,21]. Aging is accompanied with depletion of many non-enzymatic endogenous antioxidants in tissue and plasma of rodents that contribute to low oxygen radical absorbance capacity upon aging [22]. The increase in the incidences of liver diseases while aging is well accepted [23,24]. The mechanism of liver ageing is though not well understood still oxidative stress is recognized as a major possible contributor and reports are also available that suggested the induction of MAPK pathways by ROS [25]. Ageing affects the human liver which is manifested by decreased blood flow and volume along with cellular alterations like higher oxidative stress, mitochondrial dysfunction, and many more [26]. Apart from this, the ageing is also a key risk factor in numerous specific hepatic diseases including non-alcoholic fatty liver disease (NAFLD) [27]. It is interesting to note that treatments related to the liver are more often proposed to the elderly population [28]. A previous study has reported that Fenugreek extract improved lipid metabolism [29] and a clinical study has shown that antioxidant capacity can be improved by the fenugreek seeds as well [30].

In this study, we have observed that fenugreek seed dietary supplementation has a positive effect in the regulation of the hepatic enzymes in the aged mice. Elevated SOD levels by fenugreek seeds supplementation is evidence that this widely used dietary ingredient might have a potential role in the hepatoprotection in aging mediated by oxidative damage. A detailed representation is given in Figure 4. Normally all cells are equipped with endogenous defense mechanisms, such as superoxide dismutase (SOD), glutathione peroxidase (GPx), and glutathione reductase (GR) to metabolize the toxic intermediates and protect against ROS-induced damage [31]. Antioxidant enzymes are the major defense system of cells in normal aerobic reactions. It is known that a reduction in SOD activity can result in significant oxidative stress and cell death, and the antioxidant defense system must be tightly regulated under normal physiological conditions. A markedly depressed antioxidant profile and increased production of superoxide anions is seen in aged tissues in many different animals [32].

Previous studies showed that the reduced GSH is oxidized to GSSG by GPx during the catalytic cycle and recycling of GSSG by GR is important for the intracellular GSH homeostasis and GPx functionality. The aging body produces a large amount of ROS, which reduces the body’s ability to scavenge free radicals and the activity of antioxidant enzymes, causing oxidative damage to the body [33]. The SOD is a naturally occurring superoxide free radical scavenger that converts harmful superoxide radicals into hydrogen peroxide [34]. A recent study conducted by Yuan and colleagues [35] on the anti-aging, antioxidant, and organ-protective potential of *Flammulina velutipes* polysaccharide correlated severe oxidative damage with the decrease in SOD, glutathione peroxidase, and other enzymes and increase in MDA levels. Further, the results of this work revealed that the treatment with sulfated polysaccharide significantly enhanced the antioxidant enzymes and reduced lipid peroxidation products. This is in line with our study as we have found similar effects on SOD, GR, and MDA levels from the fenugreek seed supplementation.

In a study conducted to evaluate the protective effect of fenugreek seeds against ethanol toxicity, Thirunavukkarasu and coworkers found that inhibition of GPx activity by ethanol can cause H_2_O_2_ accumulation along with its decomposition products [36]. We have observed that in aged mice before supplementation, the activity of GPx and GR is higher than in mice after supplementation with fenugreek. Perhaps the high antioxidant potential of supplemented fenugreek is associated with reducing reactive oxygen species which leads to a feedback regulation loop suppressing the levels of GPx and GR. The increase in SOD activity after supplementation with fenugreek may also be related to its high free radical scavenging potential. Enhanced total phenolic contents and DPPH free radical scavenging activity clearly suggests that antioxidant defense mechanisms are improved by fenugreek seed dietary supplementation.

Whenever there is oxidative stress, there is a critical need for antioxidants. A good external source for antioxidants is food rich in polyphenols, such as fruits and vegetables [37,38]. The early study of Lopez-Alarcona and Denicola [39] and Toda [33] showed that the phytochemical analysis of fenugreek revealed the presence of a number of polyphenols which were investigated by several groups worldwide to study the antioxidant properties of fenugreek. Naidu and colleagues evaluated the extracts of fenugreek seed and showed that the seeds of fenugreek possessed significant ability to scavenge free-radicals and showed antioxidant activity [13].

Fenugreek contains a number of important, beneficial flavonoids, and polyphenol compounds. Nagulapalli Venkata et al. [34] have reported the presence of a wide range of flavonoids, namely quercetin, luteolin, vitexin, and 7, 4′-dimethoxy flavanones in the alcoholic extracts of the whole plant. The other groups have reported similar findings of the presence of aglycones kaempferol, quercetin, tricin, and naringenin [40]. The antioxidant potential of fenugreek seeds is well acknowledged. Pandey and Awasthi [41] evaluated the antioxidant activity of the extracts of three different fenugreek seed flours (soaked, germinated, and roasted) and reported that the antioxidant activity was in the range of 32–73.89% and also 18.1% antioxidant activity was exhibited by the raw fenugreek seed flour. Another study revealed significant antioxidant activity in germinated fenugreek seeds which was correlated with the presence of flavonoids and polyphenols [42].

The effects of fenugreek supplementation may be mediated through different mechanisms. While our data do not allow to clearly demonstrate the mechanism of fenugreek seeds bioeffects in ageing mice, the possible role of mitochondria cannot be omitted. The oxidative stress is due to disequilibrium between antioxidants and ROS. Evidence suggests that a subtle balance among antioxidants, especially in mitochondria, is vital in preventing ROS production. The role of various liver enzymes in mitochondrial oxidative stress is well defined [43]. Along this line, flavonoids from fenugreek seeds were reported to have a positive effect on the improvement of mitochondrial dysfunction and reported to decrease the accumulation of intracellular ROS, and also showed a protective effect on mitochondrial DNA from oxidative damage [44].

## 5. Conclusions

This study provides clear evidence about fenugreek supplementation modulatory action targeting the antioxidant defense systems in the livers of aging mice. We have found that the activities of different antioxidant liver enzymes were altered in response to fenugreek seeds supplementation. It was found that fenugreek seed dietary supplementation significantly increased SOD activity in the aged mice liver and decreased GPx and GR1 enzyme activities. Additionally, the higher levels of phenolic contents and DPPH radical scavenging properties were found upon supplementation. Taken together these data show that fenugreek seeds dietary supplement counteracts the oxidative stress in the aging mice livers. However, while many previous studies showed a suppressive effect of fenugreek seeds and oil on the level of lipid peroxidation, in our work we just observed a tendency for decrease of MDA levels, which did not reach statistical significance.

Our findings warrant further research on potential benefits of fenugreek seed dietary supplementation for the geriatric populations, and in particular substantial future clinical studies are required to prove its beneficial effect and adequate dose for human consumption.

## Figures and Tables

**Figure 1 nutrients-12-02552-f001:**
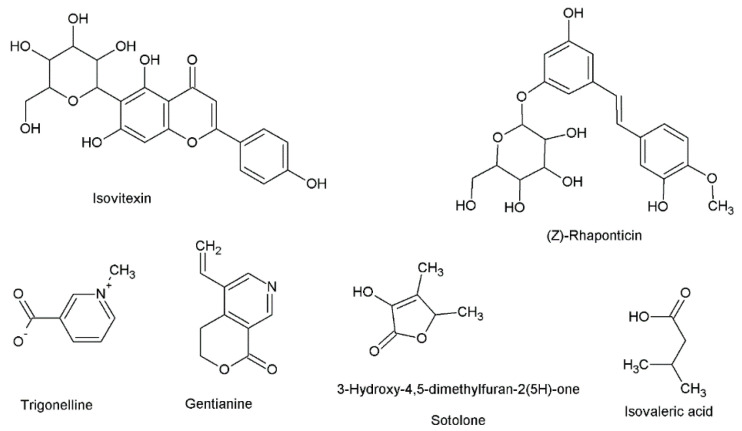
Chemical structures of some important phytoconstituents present in fenugreek seeds.

**Figure 2 nutrients-12-02552-f002:**
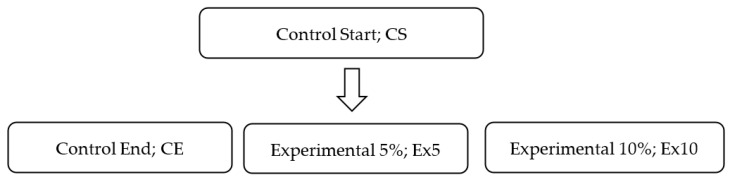
Scheme of experiment with fenugreek supplementation on 12-months old male mice.

**Figure 3 nutrients-12-02552-f003:**
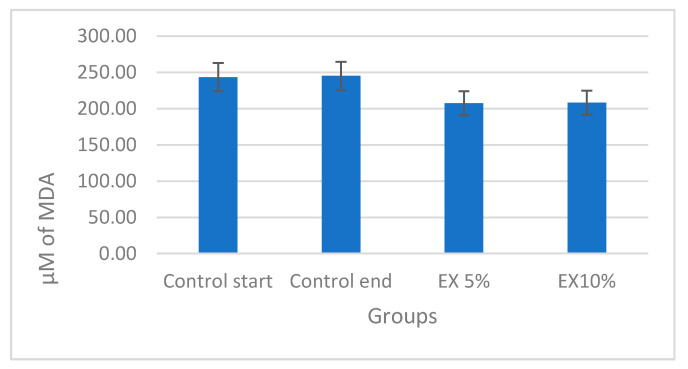
The malondialdehyde level (µM) in mice liver in estimated groups.

**Figure 4 nutrients-12-02552-f004:**
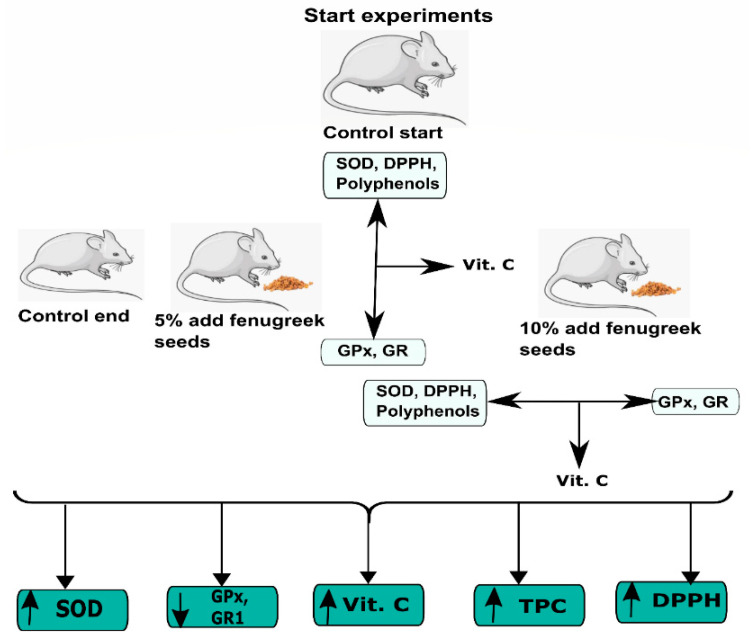
A schematic representation of the experimental procedure and the effect of fenugreek supplement on different parameters studied. Fenugreek seed supplementation decreases the activity of GPx and GR and enhanced the levels of SOD activity, Vitamin C, total phenolic content, and DPPH scavenging activity.

**Table 1 nutrients-12-02552-t001:** Chemical composition of the used feed (Altromin Spezialfutter GmbH & Co.).

Ingredient	Content (in mg/kg)
Protein	174,466
Fat	50,500
Fiber	29,980
Ash	55,886
Moisture	59,418
Vitamin C	20

**Table 2 nutrients-12-02552-t002:** The activity of estimated enzymes superoxide dismutase (SOD), glutathione peroxidase (GPx), and glutathione reductase (GR) (LSM ± SD) in mice liver.

	SOD U/mL (LSM ± SD)	GPx nmol/min/mL (LSM ± SD)	GR l nmol/min/mL (LSM ± SD)
**Control start**	21.97 ± 0.32 ^A^	11.71 ± 0.39 ^A^	165.55 ± 4.9 ^A^
**Control end**	21.91 ± 0.6 ^A^	11.93 ± 0.29 ^A^	166.81 ± 3.39 ^A^
**EX 5%**	30.64 ± 1.04 ^B^	7.15 ± 0.55 ^B^	96.58 ± 3.35 ^B^
**EX10%**	35.87 ± 1.61 ^B^	4.12 ± 0.14 ^C^	87.94 ± 1.54 ^B^

^A–C^ Columns with different letters differed significantly (*p* < 0.01).

**Table 3 nutrients-12-02552-t003:** The level of estimated antioxidant vitamin C, total polyphenols, and DPPH in mice liver.

	Vitamin C mg/100g Tissue (LSM ± SD)	Total Polyphenols mg of GAE/g Tissue (LSM ± SD)	Radical Scavenging Activity by DPPH % (LSM ± SD)
Control start	1.35 ± 0.03	1.71 ± 0.03 ^a^	43.79 ± 1.27 ^a^
Control end	1.33 ± 0.01	1.70 ± 0.02 ^a^	43.81 ± 0.39 ^a^
EX 5%	1.37 ± 0.03	1.93 ± 0.03 ^a,b^	48.64 ± 1.07 ^a,b^
EX10%	1.41 ± 0.02	2.06 ± 0.08 ^b^	51.84 ± 1.75 ^b^

^a,b^ Columns with different letters differed significantly (*p* < 0.05).

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
