# Peer review of "Fenugreek (Trigonella foenum-graecum L.) Seeds Dietary Supplementation Regulates Liver Antioxidant Defense Systems in Aging Mice"

_nutrients, 2020, doi:10.3390/nu12092552_

Round 1
Reviewer 1 Report
The investigated the effect of fenugreek seeds dietary supplementation in the aspects of antioxidant defense parameters regulation in aging mice.
The dietary solutions of organs aging delaying conducted in vivo or in vitro models are still of the interest of many researchers. So, from that point of view, the presented research is interesting.
However, the manuscript at present form is far from being accepted for publication in "Nutrients" Journal. The text is incomplete and requires a major revision!
I have some comments and suggestion for the whole text:
- The whole text must be checked for correctness and some colloquial terms must be removed.
The Introduction section:
- I my opinion, a crucial aspects about the bioactive ingredients ) of fenugreek seed (i.e. the characteristic of phenolic compounds and fatty acids composition) were omitted in the Introduction section.
The Materials and Methods:
- The in vivo experiment is presented inadequately. Please indicate the Number of Document of Approval for the animal experiment issued by The Local Ethical Commission for Animal Experiments. There was no description of animal euthanasia procedure after the experiment in the text!
- The scheme of experimental design of the animal study would increase the attractiveness the text for the readers.
- I am not convinced that the data presented at Table 1 should be given to two decimal places.
- What was the source of fenugreek seeds?
- How were the experimental feeds prepared and stored?
- Why the Authors did not mention about the statistical analyses of the results?
The results:
- The study should include at least the results of antioxidant potential of feeds use in animal study. The better option would be the bioactive compounds characteristic.
- In addition to the results presented in the text, feed intake, body weight gains, liver to body weight ratio, liver macroscopic and histopathological changes after feeding the animals with different types of experimental diets should be also included!
- The units of enzymes and DPPH were lacking.
Discussion:
- The Discussion must be re-written. At a present form is not acceptable, because it is more a review, not a comparison with the results of others studies.
- Discussion should reflect the significance of fenugreek seeds compounds in relation to oxidative defense parameters in the liver of aging mice.
The Conlusion should be a separate part of the manuscript and must be re-written based on the corrected Discussion of the results.
A graphical abstract is highly recommended.
Author Response
Answers to Reviewers' Comments
Preface.
We thank you very much for the careful revision to our manuscript. We realized from the comments received that several key points in our original work have not been properly addressed. Additionally, we also recognized and appreciated the competence of the Reviewers. We have incorporated most of the suggestions of the reviewers that certainly improved the quality of our manuscript.
In this resubmitted version, we took advantage of the criticisms received and modified the text accordingly, adding explanations and new experimental results. In the short given time, we made a significant effort adding new data to the original version. We could not do much more than this due to the turbulent situation caused by Covid-19 emergency that significantly limited our experimental activities.
Additional Fig. 1, Fig 2 and Fig 3, and a graphical abstract are now added to the manuscript. The text has been largely updated with new information and comments. Several points have been clarified. The additional data on the monoaldehyde levels was also generated and incorporated in the revised version.
Several minor corrections have been also introduced to make the text more fluent.
Below, you will find, point-by-point, our answers to the Reviewers.
All changes in the revised text are indicated with yellow highlight.
Review Report 1 of Reviewer #1 (R1)
Query 1: The investigated effect of fenugreek seeds dietary supplementation in the aspects of antioxidant defense parameters regulation in aging mice. The dietary solutions of organs aging delaying conducted in vivo or in vitro models are still of the interest of many researchers. So, from that point of view, the presented research is interesting.
Reply 1: We thank the Reviewer for the detailed assessment of our manuscript. We did our best to improve our manuscript with additional information. We have worked upon all the suggestions/comments raised by the Reviewer.
Query 2: However, the manuscript at present form is far from being accepted for publication in "Nutrients" Journal. The text is incomplete and requires a major revision! I have some comments and suggestion for the whole text:
Reply 2: We thank the Reviewer for all suggestions. We have majorly revised the manuscript and added all the suggestions and addressed all the points raised by the Reviewer.
Query 3: The whole text must be checked for correctness and some colloquial terms must be removed.
Reply 3: We revised the entire manuscript and thoroughly checked the whole text for the correctness.
Query 4: The Introduction section: I my opinion, a crucial aspect about the bioactive ingredients of fenugreek seed (i.e. the characteristic of phenolic compounds and fatty acids composition) were omitted in the Introduction section.
Reply 4: We thank the Reviewer for this important suggestion. Accordingly, we have added sufficient information about the bioactive ingredients of the fenugreek seeds. In addition to this we have added Fig 1 to present the major phytoconstituents present in fenugreek seeds (Line # 79-92 and Fig 1).
Query 5: The Materials and Methods: The in vivo experiment is presented inadequately. Please indicate the Number of Document of Approval for the animal experiment issued by The Local Ethical Commission for Animal Experiments.
Reply 5: We have indicated the Number of Document of Approval for the animal experiment issued by The Local Ethical Commission for Animal Experiments. (Line # 124-127).
Four hours after the final treatment all animals were anesthetized with diethyl ether and then decapitated. Immediately after decapitation, the liver was isolated and processed for biochemical studies.
Query 6: The scheme of experimental design of the animal study would increase the attractiveness the text for the readers.
Reply 6: We thank the Reviewer for this excellent suggestion and we have added 2 schemes (Fig 2 and 4) of the of experimental design of the animal study.
Query 7: I am not convinced that the data presented at Table 1 should be given to two decimal places.
Reply 7: We agree with the Reviewer, and the data presented in table 1 is now given as whole units (mg/kg).
Query 8: What was the source of fenugreek seeds? How were the experimental feeds prepared and stored?
Reply 8: The fenugreek seed were purchased from the local pharmacy Herbapol in Cracow SA, Polish herbs company (Nr. of claims GIS-HŻ-4433-D-168/AW/05, GTIN: 5903850001270). The seeds were kept in the original manufacturer's packaging in a refrigerator (in temperature 5 °C +/- 1.5°C). The respective information is now added in the manuscript (Line # 114-116).
Query 9: Why the Authors did not mention about the statistical analyses of the results?
Reply 9: We are sorry for this inadvertent error, and the detailed description of the statistical analysis is now added in the revised manuscript (Line # 210-219).
Query 10: The results: The study should include at least the results of antioxidant potential of feeds use in animal study. The better option would be the bioactive compounds characteristic.
Reply 10: Numerous previous studies reported on the antioxidant characteristics and bioactive compound profiles on fenugreek seeds. We have now incorporated in the revised manuscript the results reported from the earlier studies (Line # 321-336).
Query 11: In addition to the results presented in the text, feed intake, body weight gains, liver to body weight ratio, liver macroscopic and histopathological changes after feeding the animals with different types of experimental diets should be also included!
Reply 11: We thank the Reviewer for this suggestion. We have body weight and liver weight, but we did not include these data because they were not statistically significant as well as feed intake.
We did not observe any differences in the body and liver weight of the mice of different groups. The respective data are as follows:
Weight of body
CS – 33,7 +/-2,3
CE – 34,1 +/-1,9
Ex5 – 34,2 +/-2,4
Ex10 – 33,9 +/-2,1
Weight of liver
CS – 1,83 +/-0,19
CE – 1,81 +/-0,21
Ex5 – 1,82 +/-0,24
Ex10 – 1,79 +/-0,19
We also did not observe any differences in feed consumption in control (CS and CE) and experimental groups (Ex5 and Ex 10). The respective data are as follows:
The animals were kept in boxes of n=3 mice;
the daily consumption (g per box) from the cage was
CS – 6,92 +/-0,23
CE – 6,89 +/-0,17
Ex5 – 6,93 +/-0,21
Ex10 – 6,91 +/-0,21
We also did not observe any histopathological or morphological changes in the livers of the mice of the different groups.
Query 12: The units of enzymes and DPPH were lacking.
Reply 12: The units of enzymes and DPPH are now added.
Query 13: Discussion: The Discussion must be re-written. At a present form is not acceptable, because it is more a review, not a comparison with the results of others studies.
Reply 13: We have substantially revised the discussion section and compare our data with many similar studies (Line # 265-268, 283-290, 291-303, 313-346).
Query 14: Discussion should reflect the significance of fenugreek seeds compounds in relation to oxidative defense parameters in the liver of aging mice.
Reply 14: We have mentioned about the significance of the fenugreek seeds in relation to the oxidative defense parameters as mentioned in Reply 13.
Query 15: The Conlusion should be a separate part of the manuscript and must be re-written based on the corrected Discussion of the results.
Reply 15: We thank the reviewer for this important suggestion and added a separate section of conclusion based on the revised discussion. (Line # 349-363)
Query 16: A graphical abstract is highly recommended.
Reply 16: We have added a separate graphical abstract
Reviewer 2 Report
In the present study the researchers focused the attention on the possible health beneficial effects of the Fenugreek oral supplementation in a model of aging mice. In order to assess it they gave particular emphasis to the possible role of the Fenugreek as enhancer of the liver antioxidant defenses because of its apparent role in restoring several antioxidant enzymes activities like superoxide dismutase, glutathione reductase and glutathione peroxidase and by a direct scavenger biological function.
However, I have some major concerns regarding this manuscript:
- The most important one is the lack of a mouse model of liver damage. Due to the supposed antioxidant effect, as surely discussed in the preliminary study design phase, what is the reason why the authors decided to evaluate the helpful effect in “aging mice”? aging doesn’t mean inevitably a worsen of redox balance (independently to the OLD cited paper of the ref. n.7 (2001)=there are a lot of independent variables that could affect the oxidative balance and then the results of the study). There are some other most promising animal models of oxidative liver damage (alcoholic, non-alcoholic, ischemia-reperfusion etc.) in which it could be possible to obtain some more fascinating scientific prospective by the analysis of the results
- The assessment of superoxide dismutase, glutathione reductase and glutathione peroxidase activities alone is an old way to evaluate the “healthy of the antioxidant cellular systems” it would have been useful to focus the attention also on other marker of lipid peroxidation like Thiobarbituric acid reactive substances assessment or oxidative adducts (HNE, MDA), etc.
- The paper lacks an explanation of the biological mechanisms surrounding the antioxidant power of Is it a mitochondrial matter? Could be…
- Statistical analysis section was missing (sample size calculation, statistical methods description etc.)
Author Response
Manuscript Ref. No: nutrients-875109
Fenugreek (Trigonella foenum-graecum L.) seeds dietary supplementation regulates liver antioxidant defense systems in aging mice
Devesh Tewari, Artur Jóźwik, Małgorzata Łysek-Gładysińska, Weronika Grzybek, Wioletta Adamus-Białek, Jacek Bicki, Nina Strzałkowska, Agnieszka Kamińska, Olaf K. Horbańczuk, and Atanas G. Atanasov
Answers to Reviewers' Comments
Preface.
We thank you very much for the careful revision to our manuscript. We realized from the comments received that several key points in our original work have not been properly addressed. Additionally, we also recognized and appreciated the competence of the Reviewers. We have incorporated most of the suggestions of the reviewers that certainly improved the quality of our manuscript.
In this resubmitted version, we took advantage of the criticisms received and modified the text accordingly, adding explanations and new experimental results. In the short given time, we made a significant effort adding new data to the original version. We could not do much more than this due to the turbulent situation caused by Covid-19 emergency that significantly limited our experimental activities.
Additional Fig. 1, Fig 2 and Fig 3, and a graphical abstract are now added to the manuscript. The text has been largely updated with new information and comments. Several points have been clarified. The additional data on the monoaldehyde levels was also generated and incorporated in the revised version.
Several minor corrections have been also introduced to make the text more fluent.
Below, you will find, point-by-point, our answers to the Reviewers.
All changes in the revised text are indicated with yellow highlight.
Review Report 1 of Reviewer 2
Query 1: In the present study the researchers focused the attention on the possible health beneficial effects of the Fenugreek oral supplementation in a model of aging mice. In order to assess it they gave particular emphasis to the possible role of the Fenugreek as enhancer of the liver antioxidant defenses because of its apparent role in restoring several antioxidant enzymes activities like superoxide dismutase, glutathione reductase and glutathione peroxidase and by a direct scavenger biological function.
Reply 1: We thank the Reviewer for the detailed assessment of our manuscript. We did our best to improve our manuscript with additional information. We have worked upon all the suggestions/comments raised by the reviewer.
Query 2: The most important one is the lack of a mouse model of liver damage. Due to the supposed antioxidant effect, as surely discussed in the preliminary study design phase, what is the reason why the authors decided to evaluate the helpful effect in “aging mice”? aging doesn’t mean inevitably a worsen of redox balance (independently to the OLD cited paper of the ref. n.7 (2001)=there are a lot of independent variables that could affect the oxidative balance and then the results of the study). There are some other most promising animal models of oxidative liver damage (alcoholic, non-alcoholic, ischemia-reperfusion etc.) in which it could be possible to obtain some more fascinating scientific prospective by the analysis of the results
Reply 2: We thank the Reviewer for his/her insightful comment. We have decided to evaluate the bioeffect of fenugreek seed supplement in ageing mice due to several reasons: The first and foremost reason was to evaluate the effect of fenugreek seeds in the aging mice as the fenugreek seeds are very well known and widely preferred for the geriatric populations in Asian countries and therefore the aged mice are a suitable model to conduct relevant preclinical investigation. Other previous studies also used old mice to investigate the effect of different plant extracts on lipid peroxidation and antioxidant status in male mice as a function of age. For instance, Sai et al., 2015 evaluated effect of Potentilla fulgens extract on lipid peroxidation and antioxidant status in male mice as a function of age. Although, as mentioned by the Reviewer “aging doesn’t mean inevitably a worsen of redox balance” however, it is also important to note that the theory of association of free radicals with aging received emerging acceptance as a plausible explanation of the chemical reactions for ageing process. Therefore, we believe that the correlation of aging process and free radicals can not be ignored.
Additionally, we agree with the reviewer that other animal models of oxidative liver damage (alcoholic, non-alcoholic, ischemia-reperfusion etc.) are also available and therefore in future studies we will try to design experiments to understand more about the mechanism of the fenugreek supplementation using other methods as well.
Query 3: The assessment of superoxide dismutase, glutathione reductase and glutathione peroxidase activities alone is an old way to evaluate the “healthy of the antioxidant cellular systems” it would have been useful to focus the attention also on other marker of lipid peroxidation like Thiobarbituric acid reactive substances assessment or oxidative adducts (HNE, MDA), etc.
Reply 3: We thank the reviewer for this comment. However, due to the current situation of Covid 19 pandemic extensive experimental activities are restricted. Nevertheless, we tried our best to generate some additional data as suggested by the Reviewer and additional MDA tests are now added to the revised manuscript.
Query 4: The paper lacks an explanation of the biological mechanisms surrounding the antioxidant power of Is it a mitochondrial matter? Could be…
Reply 4: We thank the Reviewer for this important suggestion and accordingly we have added a separate paragraph to state the plausible biological mechanism surrounding the antioxidant relevance of the mitochondrial matter (Line # 337-346)
Query 5: Statistical analysis section was missing (sample size calculation, statistical methods description etc.)
Reply 5: We are sorry for this inadvertent error and the detailed description of the statistical analysis is added in the revised manuscript (Line # 210-219).
Additionally,
- The reference list has been modified as we have added several new references. Special attention is given to conform to the order of references and bibliographic style of the journal.
- The entire manuscript has been thoroughly checked and edited to ensure uniform style, organization and quality.
On behalf of my co-authors, I once again express my sincere thanks to the erudite reviewers for the valuable suggestions and constructive input to improve the quality of our manuscript.
Round 2
Reviewer 1 Report
Comments for the Authors (revised manuscript)
The Authors mostly addressed my suggestions, but some issues remain to be explained and corrected.
1. The procedure of animals euthanasia described in answers to reviewer raises my doubts. I don’t know, why did the Authors used diethyl ether? This substance is not used in animal experiments since 20 years, because of serious side effects to experimenters and is irritating to animals. Moreover, diethyl ether is frequently prohibited as the substance used for anesthesia and may influence i.e. on liver enzymes activities. The authors must prove that they had permission from the relevant Ethics Commission for Animal Experiments and Care for use the diethyl ether for animal anesthesia! The information of the procedure of animal euthanasia should be added to the text!
2. The information, what was the average initial weight of the animals at the beginning of experiment should be provided! The same, the information about the source of fenugreek seeds, the final body weights (or absolute changes of body weights, liver weights, feed intake and liver histopathological assessment should be provided in manuscript.
3. Some typos are still present in the text.
Author Response
Dear Reviewer
Manuscript Ref. No: nutrients-875109
Fenugreek (Trigonella foenum-graecum L.) seeds dietary supplementation regulates liver antioxidant defense systems in aging mice
Devesh Tewari, Artur Jóźwik, Małgorzata Łysek-Gładysińska, Weronika Grzybek, Wioletta Adamus-Białek, Jacek Bicki, Nina Strzałkowska, Agnieszka Kamińska, Olaf K. Horbańczuk, and Atanas G. Atanasov
Answers to Reviewers' Comments
We thank you very much for the careful review to our revised manuscript.
Review Report of Reviewer #1 (R1)
Query 1: The Authors mostly addressed my suggestions, but some issues remain to be explained and corrected.
Reply 1: We thank the Reviewer for the detailed assessment of our revised manuscript. We did our best to address all the remaining issues raised by the reviewer.
Query 2: The procedure of animals euthanasia described in answers to reviewer raises my doubts. I don’t know, why did the Authors used diethyl ether? This substance is not used in animal experiments since 20 years, because of serious side effects to experimenters and is irritating to animals. Moreover, diethyl ether is frequently prohibited as the substance used for anesthesia and may influence i.e. on liver enzymes activities. The authors must prove that they had permission from the relevant Ethics Commission for Animal Experiments and Care for use the diethyl ether for animal anesthesia! The information of the procedure of animal euthanasia should be added to the text!
Reply 2: We agree with the comment of the Reviewer. We want to clarify that all operation took place in laboratory hood in special cages (UNO company, the Netherlands) where we just briefly dosed the animals with diethyl ether manually so as to incapacitate them. The next step was decapitation because we also took blood from the mice by bleeding. We agree that this method is old but it gives about 1.5 mL of blood per mice and we can use fewer animals in the experiment. Additionally, the experiment protocol was approved by the ethical committee (approval number No. 46/2016). Both the “control” and the “treatment” groups were subjected to the same handling – so differences between them cannot be due to euthanasia procedures. This information is now included on page 4 of the revised manuscript.
Query 3: The information, what was the average initial weight of the animals at the beginning of experiment should be provided! The same, the information about the source of fenugreek seeds, the final body weights (or absolute changes of body weights, liver weights, feed intake and liver histopathological assessment should be provided in manuscript.
Reply 3: We thank the Reviewer for this comment. On page 4 of the revised manuscript we now provide information for average food intake, for the weights of the animals of the different groups (including average initial and final weight), as well as for liver histopathological assessments and average liver weights of the animals of the different groups. On page 3 of the revised manuscript, information for the source of fenugreek seeds is also provided.
Query 4: Some typos are still present in the text.
Reply 4: We corrected the typos present in the manuscript.
Kind regards,
Artur Jóźwik
Reviewer 2 Report
Reply 2: We thank the Reviewer for his/her insightful comment. We have decided to evaluate the bioeffect of fenugreek seed supplement in ageing mice due to several reasons: The first and foremost reason was to evaluate the effect of fenugreek seeds in the aging mice as the fenugreek seeds are very well known and widely preferred for the geriatric populations in Asian countries and therefore the aged mice are a suitable model to conduct relevant preclinical investigation. Other previous studies also used old mice to investigate the effect of different plant extracts on lipid peroxidation and antioxidant status in male mice as a function of age. For instance, Sai et al., 2015 evaluated effect of Potentilla fulgens extract on lipid peroxidation and antioxidant status in male mice as a function of age. Although, as mentioned by the Reviewer “aging doesn’t mean inevitably a worsen of redox balance” however, it is also important to note that the theory of association of free radicals with aging received emerging acceptance as a plausible explanation of the chemical reactions for ageing process. Therefore, we believe that the correlation of aging process and free radicals can not be ignored.
Additionally, we agree with the reviewer that other animal models of oxidative liver damage (alcoholic, non-alcoholic, ischemia-reperfusion etc.) are also available and therefore in future studies we will try to design experiments to understand more about the mechanism of the fenugreek supplementation using other methods as well.
- As previously reported the most important concern regarding this paper is the risk to under or overestimate the results of a study based on a barely used model of (SYSTEMIC) oxidative stress damage, is too high. Also focusing the attention on the results of MDA assessment, the obtained outcome could be linked to a not perfectly adherent methodology regarding the established endpoint. In other words: the model is not fitting with your aim. The authors probably referred their reply to SaiO V et al. Redox Rep, 2016… as previously exposed it doesn’t mean that the used model is a GOOD model of Liver oxidative stress induced damage (1 paper in comparison to 1000 concerning other models is a too big difference), in particular also considering the IF of the cited Journal.
Reply 3: We thank the reviewer for this comment. However, due to the current situation of Covid 19 pandemic extensive experimental activities are restricted. Nevertheless, we tried our best to generate some additional data as suggested by the Reviewer and additional MDA tests are now added to the revised manuscript.
- I’m so sorry for this answer. The authors must consider the possibility to resubmit the paper to this journal or to another one when they will be able to examine in depth the topic considering also the immunometabolism concept and the involvement of the electronic transport chain in this context.
Reply 4: We thank the Reviewer for this important suggestion and accordingly we have added a separate paragraph to state the plausible biological mechanism surrounding the antioxidant relevance of the mitochondrial matter (Line # 337-346)
- See the previous comment
Reply 5: We are sorry for this inadvertent error and the detailed description of the statistical analysis is added in the revised manuscript (Line # 210-219).
- Sample size calculation is still missing
Author Response
Dear Reviewer
Manuscript Ref. No: nutrients-875109
Fenugreek (Trigonella foenum-graecum L.) seeds dietary supplementation regulates liver antioxidant defense systems in aging mice
Devesh Tewari, Artur Jóźwik, Małgorzata Łysek-Gładysińska, Weronika Grzybek, Wioletta Adamus-Białek, Jacek Bicki, Nina Strzałkowska, Agnieszka Kamińska, Olaf K. Horbańczuk, and Atanas G. Atanasov
Answers to Reviewers' Comments
We thank you very much for the careful review to our revised manuscript.
Review Report of Reviewer 2
Query 1: - As previously reported the most important concern regarding this paper is the risk to under or overestimate the results of a study based on a barely used model of (SYSTEMIC) oxidative stress damage, is too high. Also focusing the attention on the results of MDA assessment, the obtained outcome could be linked to a not perfectly adherent methodology regarding the established endpoint. In other words: the model is not fitting with your aim. The authors probably referred their reply to SaiO V et al. Redox Rep, 2016… as previously exposed it doesn’t mean that the used model is a GOOD model of Liver oxidative stress induced damage (1 paper in comparison to 1000 concerning other models is a too big difference), in particular also considering the IF of the cited Journal.
Reply 1: We thank the Reviewer for the detailed assessment of our revised manuscript. Enzymes like SOD, GPx and GR are mostly investigated to understand hepatoprotective effects not only in ageing mice but also in a broad variety of other models, including human. Apart from that, there is a clear association of antioxidant enzymes with ageing, e.g., Veal E., Jackson T., Latimer H. (2018) Role/s of ‘Antioxidant’ Enzymes in Ageing. In: Harris J., Korolchuk V. (eds) Biochemistry and Cell Biology of Ageing: Part I Biomedical Science. Subcellular Biochemistry, vol 90. Springer, Singapore. https://doi.org/10.1007/978-981-13-2835-0_14. Therefore, we believe that our data are of relevance and would be of interest for the scientific community.
Query 2: Reply 3: We thank the reviewer for this comment. However, due to the current situation of Covid 19 pandemic extensive experimental activities are restricted. Nevertheless, we tried our best to generate some additional data as suggested by the Reviewer and additional MDA tests are now added to the revised manuscript.
- I’m so sorry for this answer. The authors must consider the possibility to resubmit the paper to this journal or to another one when they will be able to examine in depth the topic considering also the immunometabolism concept and the involvement of the electronic transport chain in this context.
Reply 2: We did our best to improve the manuscript in the existing extraordinary situation. We do ask the Reviewer to have understanding. Such more detailed studies are not feasible within the current study, but we do agree with the Reviewer that this could be a highly interesting topic for future investigation.
Query 3: - Sample size calculation is still missing
Reply 3: We thank the Reviewer for pointing out this omission and for the improvement recommendation. Since power analysis was not possible, we determined the needed sample size according to the resource equation approach, using the free web tool available at http://wnarifin.github.io. This information is now provided on page 4 of the revised manuscript.
Kind regards,
Artur Jóźwik